# CONVOLUTIONAL NEURAL NETWORK DYNAMICS: A GRAPH PERSPECTIVE

## ABSTRACT

The success of neural networks (NNs) in a wide range of applications has led to increased interest in understanding the underlying learning dynamics of these models. In this paper, we go beyond mere descriptions of the learning dynamics by taking a graph perspective and investigating the relationship between the graph structure of NNs and their performance. Specifically, we propose (1) representing the neural network learning process as a time-evolving graph (i.e., a series of static graph snapshots over epochs), (2) capturing the structural changes of the NN during the training phase in a simple temporal summary, and (3) leveraging the structural summary to predict the accuracy of the underlying NN in a classification or regression task. For the dynamic graph representation of NNs, we explore structural representations for fully-connected and convolutional layers, which are key components of powerful NN models. Our analysis shows that a simple summary of graph statistics, such as weighted degree and eigenvector centrality, over just a few epochs can be used to accurately predict the performance of NNs. For example, a weighted degree-based summary of the time-evolving graph that is constructed based on 5 training epochs of the LeNet architecture achieves classification accuracy of over 93%. Our findings are consistent for different NN architectures, including LeNet, VGG, AlexNet, and ResNet.

## 1 INTRODUCTION

Neural networks (NNs) have driven advancements in many domains, including computer vision and image processing (Hu et al., 2018), natural language processing (Sutskever et al., 2014; Bahdanau et al., 2015), and bioinformatics (Cao et al., 2020; Li et al., 2019). As task complexity increases, networks grow deeper and larger, consequently requiring more computational resources and training data, as well as sacrificing interpretability for improved task performance. Some works have focused on understanding and interpreting deep NNs (Raghu et al., 2017; Chakraborty & et al., 2017; Ioffe & Szegedy, 2015; He et al., 2016). One approach towards this goal involves representing the NN as its underlying graph structure, and studying selected graph properties, such as clustering coefficient, path length (You et al., 2020), modularity (Filan et al., 2021), persistence (Rieck et al., 2019). For example, You et al. (2020) represent NNs as relational graphs capturing the message passing process, and investigate the correlation between the predictive performance of NNs and architectural changes. However, the studies of NN structures as graphs are limited, and the structural changes of the underlying graph during the training process have been largely overlooked in the literature.

To fill this gap, in this work, we take the graph perspective and aim to predict the performance of an NN by capturing early NN dynamics during the training phase. Successful performance prediction based on only a few epochs could be used for early stopping (Yu & Zhu, 2020), and thus, more efficient NN training. To solve the performance prediction problem, we propose a multi-step framework, depicted in Fig. 1. Specifically, we propose to represent the underlying graph structure of an NN as a time-evolving k-partite graph, where each part corresponds to a different NN layer, and each graph snapshot in the evolving graph maps to an NN instance at a specific epoch. We build on existing graph representations of fully-connected and convolutional layers, both of which are key components of popular NN architectures, and introduce a new, compact, efficient-to-compute ("rolled") graph representation for convolutional layers. Then, we extract well-known node features (weighted degree and eigenvector centrality) from the time-evolving graph and construct temporal signatures by computing summary statistics on the node feature distributions. Finally, we cast NN

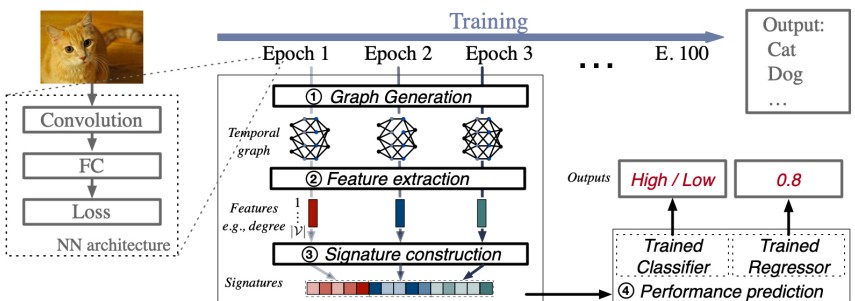

Figure 1: Our proposed framework for predicting NN performance in a downstream image classification task, shown for one test instance. The input to our framework is an NN trained for a few epochs (3, in this example). Steps: (S1) the input NN is converted to three static graphs, each representing one training epoch; (S2) node features (e.g., degree) are extracted from each graph snapshot (i.e., one per epoch); (S3) in order to summarize the changes in the graph structure over time, a signature vector is constructed by aggregating the node features per graph snapshot and concatenating the individual snapshot signatures; (S4) a pre-trained classifier and regressor predict the performance of the input NN given the signature vector from (S3).

performance prediction as a classification task and a regression task, each of which operates on the temporal structural signatures of the NN. Our main contributions are:

- **New graph-based NN representation:** We introduce a new graph representation for convolutional layers that is more compact and efficient-to-compute than the existing unrolled representation (Rieck et al., 2019), while not sacrificing accuracy in the NN performance prediction task.
- **Framework for NN performance prediction:** We propose a simple, multi-step graph-based framework to solve the NN performance prediction problem by capturing the early NN dynamics during the training phase.
- **Extensive empirical analysis:** Using well-known image classification datasets (ImageNet and CIFAR-10), and a variety of NN architectures (AlexNet, VGG, LeNet, and ResNet), we show that our framework can effectively predict the performance of NNs by observing only a few epochs of training, well before their corresponding early stopping epochs. For instance, using our framework to capture the changes in the graph structure in only 5 training epochs of the ResNet architecture results in classification accuracy of over 90%.

## 2 PRELIMINARIES

We first present the key concepts that our work builds upon. Table 1 gives the major symbols and their descriptions.

An NN model is a collection of connected units (neurons) that are organized in layers, and is defined by a set of parameters that adjust during the training process. We refer to the training process of a single architecture along with its respective hyperparameters as an 'instance'. We focus on two types of layers that are the key components of many powerful NN models such as LeNet(Lecun et al., 1998), VGG (Simonyan & Zisserman, 2015) and ResNet(He et al., 2016): fully connected layers (fc) and convolutional (conv) layers.

### 2.1 GRAPHS: TERMINOLOGY AND NOTATION

Let $\mathbf{G} = (\mathcal{V}, \mathcal{E})$ be an undirected, weighted graph with node set $\mathcal{V}$, edge set $\mathcal{E}$, and weighted adjacency matrix $\mathbf{W} \in \mathbb{R}^{|\mathcal{V}| \times |\mathcal{V}|}$. The neighbors of node $v$ are defined as $\mathcal{N}_G(v) = \{u : (u, v) \in \mathcal{E}\}$; i.e., the set of all nodes that connect directly to $v$. Graph $\mathbf{G}$ is $k$-partite if its nodeset $\mathcal{V}$ can be partitioned into $k$ independent sets: $\mathcal{V} = \bigcup_{i=1}^{k} \mathcal{V}_i$ and $\mathcal{V}_i \cap \mathcal{V}_j = \emptyset, i \neq j$. A time-evolving graph is a series of static graph snapshots over time: $\mathcal{G} = \{\mathbf{G}^1, \mathbf{G}^2, ..., \mathbf{G}^T\}$. The graphs in the series may have different nodesets and edgesets.

Here we focus on two of the most commonly-used node features in graph mining and network science, degree and eigenvector centrality, which capture different types of node importance or influence. We discuss the interpretation of these features in the context of neural network dynamics in § 3.2.

- **Degree**: Degree is the simplest and most efficient to compute node feature, and captures the connectivity of a node. The weighted degree of node $v$ is defined as the sum of weights of the edges that are incident to $i$: $d_i^w = \sum_j \mathbf{W}_{ij}$. Tracking changes in the degree of a node over time is the most direct way of capturing its structural evolution.

- **Eigenvector centrality**: This centrality is a sophisticated extension of degree centrality, related to Google's PageRank, which indicates the influence of a node in $\mathbf{G}$ (Bonacich, 1972). If a node is connected to several nodes with high eigenvector centrality, then that node will have high centrality. The eigenvector centrality of node $i$ is defined as the $i^{th}$ element in the principal eigenvector $\mathbf{v}$ of $\mathbf{W}$ (i.e., the eigenvector that corresponds to the largest eigenvalue $\lambda$): $\mathbf{W}\mathbf{v} = \lambda_{max}\mathbf{v}$.

## 2.2 FULLY-CONNECTED LAYERS: GRAPH REPRESENTATION

Let $\mathbf{x} \in \mathbb{R}^{D_{in}}$ be the input vector, $\mathbf{W} \in \mathbb{R}^{D_{in} \times D_{out}}$ the learnable weight matrix, $\mathbf{b} \in \mathbb{R}^{D_{out}}$ the bias vector and $z$ a non-linear activation functions. Then, the output of this layer is given as $\mathbf{y} = z(\mathbf{x}\mathbf{W} + \mathbf{b})$, where $D_{in}$ and $D_{out}$ correspond to the dimension of input and output layers. Fully-connected layers are straightforward to represent with a weighted, undirected graph (Filan et al., 2021). Each neuron, including those in the input and output layers, corresponds to a node. Two neurons are connected via an edge if they appear in consecutive layers. More formally, let nodeset $\mathcal{V}_i$ be the set of all neurons at

Table 1: Major symbols and definitions

| Symbol | Definition |
|---|---|
| $\mathbf{G} = (\mathcal{V}, \mathcal{E})$ | a graph with its nodeset and edgeset |
| $\mathcal{G} = \{\mathbf{G}^1, \cdots, \mathbf{G}^T\}$ | a time-evolving graph, i.e., series of graph snapshots |
| $d_v, d_v^w$ | degree and weighted degree of node $v$ |
| $\mathbf{W}$ | learnable weight matrix in an NN, and the weighted adj matrix of $\mathbf{G}$ |
| $\mathcal{K}$ | convolutional kernel |
| $D_{in/out}$ | dimension of the input/output vectors |
| $c, f$ | number of channels and filters |
| $h, w$ | height and width of an input image |
| $h_{ker}, w_{ker}$ | height and width of a kernel |
| $\mathcal{X}$ | input tensor of conv layer |

layer $i$, and nodeset $\mathcal{V}_j$ the set of all neurons at layer $j$ (which follows layer $i$). The edges that connect all nodes across the two nodesets are defined by the learnable weight matrix $\mathbf{W} \in \mathbb{R}^{|\mathcal{V}_i| \times |\mathcal{V}_j|}$.

## 2.3 CONVOLUTIONAL LAYERS: UNROLLED GRAPH REPRESENTATION

Let $\mathcal{X} \in \mathbb{R}^{c \times h \times w}$ be the input and $\mathcal{K} \in \mathbb{R}^{f \times c \times h_{ker} \times w_{ker}}$ the convolutional kernel, where $c$ is the number of channels, and $h_{ker}, w_{ker}$ are the height and width of the kernel respectively. After performing the convolution, we have $\mathcal{Y} \in \mathbb{R}^{F \times h_{out} \times w_{out}}$, where $h_{out} = h - h_{ker} + 1$ and $w_{out} = w - w_{ker} + 1$. Typical multilayer CNNs consist of convolutional layers followed by fully-connected layers. The representation of fully connected layers is straightforward but cannot be used to model

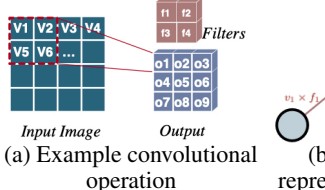 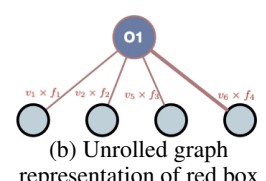

(a) Example convolutional operation / (b) Unrolled graph representation of red box

Figure 2: Unrolled graph representation example. The two typed nodes in the resultant bipartite graph map to the filtration operation and the output. For the stride in this example, the output node $o_1$ (in dark blue) is the output of 4 operation-typed nodes (in light blue).

convolutional layers. Rieck et al. (2019) proposed to "unroll" the convolution to convert conv layers into graphs. In Sec. 3, we introduce a new, compact, rolled representation that is both efficient to compute and effective at predicting the NN performance, as we show empirically in Sec. 4.

For a convolutional layer, we first unroll the convolutional operation and then represent the graph as in the case of a fully-connected layer. In this representation, the nodes and edges of the graph are defined through the convolutional operation as matrix multiplication (Gebhart et al., 2019). Specifically, for an input image $\mathcal{X}$ and kernel $\mathcal{K}_i \in \mathbb{R}^{f_i \times c_i \times h_i \times w_i}$, each node in layer $l$ is defined to be the output of the mapped feature of that input for each filter $f_i$. Edges connect each of these nodes to the corresponding nodes of the output neuron in the next layer. These edges are weighted by the activation value of that neuron (i.e., the input image for a specific stride is multiplied by the filter value at that location in the image). For an input image filter size ($c_i \times h_i \times w_i$) that results in feature-map (output) of size $1 \times o_h \times o_w$, the number of nodes of output layer of the graph representation is a function of $o_h \times o_w$, while the number of nodes in the conv operation layer of graph is a function of $c_i \times h_i \times w_i$.

As an example, Fig. 2 depicts the unrolled graph representation of a toy convolutional layer. It shows one conv operation on a small 2-dimensional input image ($4 \times 4$) and filter ($2 \times 2$). The nodes and edges of the represented graph are shown based on one stride (of size 2) of convolution operation.

# 3 NEURAL NETWORK PERFORMANCE PREDICTION: A TEMPORAL GRAPH-BASED APPROACH

In this section, we first formally introduce the problem that we seek to solve. Then, we present a new, compact, and efficient-to-compute graph representation for convolutional layers (§ 3.1), and describe our proposed temporal graph-based framework that captures the NN dynamics during training (§ 3.2).

**Problem 1 (NN Performance Prediction)** *Let $\mathcal{N} = \{N_{tr_1}, N_{tr_2}, \ldots, N_{tr_n}\}$ be a training set of $n$ NNs trained for $T$ epochs and $\mathcal{A} = \{\alpha_1, \alpha_2, \ldots, \alpha_n\}$ their corresponding downstream task accuracies (e.g., for image classification). We seek to predict the accuracy $\alpha_{tst}$ of a new instance $N_{tst}$ trained for a very small number of $t \ll T$ epochs by using $t$ epochs for the trained NNs in $\mathcal{N}$.*

Our proposed solution takes the graph perspective, and its first step is to represent each NN as a temporal graph. In addition to the graph representations that we presented in Sec. 2, we introduce a new, efficient representation for convolutional layers, which we describe next.

## 3.1 ROLLED GRAPH REPRESENTATION FOR CONVOLUTIONAL LAYERS

We call our proposed graph representation for conv layers "rolled," since it avoids unrolling the convolutional operations introduced in (Rieck et al., 2019).

**Overview & Motivation.** To preserve the semantic meaning of conv layers, we represent *each filter as a node*, and link filters in consecutive layers via weighted edges (we define the weights below), as shown in Fig. 3. The motivation behind this approach is that for larger networks, there is an explosion of nodes by unrolling the convolutions (unrolled representation explained in § 2.3), and the integrity of a unit (a single filter or kernel) becomes untenable. Also, by mapping the nodes to specific entities of NNs, our proposed graph model is more interpretable than the unrolled model, and thus it is easier to interpret the outputs of downstream graph analysis on our graph representation (e.g., computing node features, tracking the evolution of the graph).

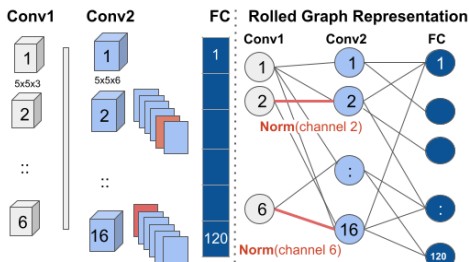

Figure 3: Rolled graph representation example. The resultant graph is a tri-partite graph with three node types (Conv1, Conv2, FC) corresponding to two convolutional layers and one FC layer. Gray and light blue nodes represent *filters* in the conv layers and dark blue nodes represent neurons in the FC layer. The red edge between nodes 6 (gray) and 16 (light blue) is weighted by the Norm(6[th] channel of filter 16). Due to dropout, the *maximum* number of edges is $6 \times 16$ between the conv layers, and $16 \times 120$ between the Conv2 and FC layers.

**Detailed Description.** Formally, let tensor $\mathcal{K}_i$ be a kernel in layer $i$ with $f_i$ filters, each with $c_i$ channels and dimensions $h_i \times w_i$. We use bracket notation to index into the kernel: for example, $\mathcal{K}_i[l, :, :, :]$ indexes the $l^{\text{th}}$ filter of kernel $\mathcal{K}_i$. We create $f_i$ nodes representing each filter $\{v_1^{(i)}, v_2^{(i)}, \ldots v_{f_i}^{(i)}\}$, with features defined as the corresponding biases in that layer. Let $\mathcal{K}_j$ be the next convolutional layer defined analogously. While edge weights between neurons in FC layers are defined in the standard way, we define the edge weights between conv layers as the norm over each kernel's channels. The edge between node $v_k^{(i)}$ representing the $k^{th}$ filter in layer $i$ (i.e., $\mathcal{K}_i[k, :, :, :]$) and node/filter $v_l^{(j)}$ in layer $j$ (i.e. $\mathcal{K}_j[l, :, :, :]$) has weight $w_{v_k^{(i)}, v_l^{(j)}} = \text{norm}(\mathcal{K}_j[l, k, :, :])$, which is the norm of the $k^{\text{th}}$ channel of the $l^{\text{th}}$ filter in the $j^{\text{th}}$ layer. Though the proposed representations enable alternative edge weight configurations, we focus on the norm of filters as other studies, including those on pruning NNs (Li et al., 2016), have demonstrated that this metric has strong correlation with filter importance. In the case of two conv layers, the resultant graph is an attributed bipartite graph with $f_i + f_j$ nodes and $f_i \times f_j$ edges, where node attributes include flattened weight vectors or filter maps. Other information such as average gradients or the bias vector can also be used for node features.

## 3.2 PROPOSED TEMPORAL GRAPH-BASED FRAMEWORK

A key objective of our work is to test whether the introduced graph representation of NNs is informative for predicting the performance of NNs. Next, we describe the steps of our proposed framework for solving Problem 1, namely performance prediction from the NN training dynamics.

As shown in Fig. 1, our method consists of four steps: **(S1)** generation of a temporal graph for the training phase of each NN; **(S2)** extraction of node features; **(S3)** construction of a feature-based graph signature that captures the NN dynamics; **(S4)** prediction of NN performance by training a classifier or regressor on the constructed signatures. We describe these steps in more detail next.

**(S1)** *Graph generation.* The first step involves converting the training process of each input NN into a time-evolving graph. Each NN $N_{tr_i} \in \mathcal{N}$, is represented as a series of checkpoints saved for $t$ training epochs. Based on these checkpoints and the graph representation approaches for fc and conv layers presented in 2.2, 3.1 and 2.3, we first convert the $\tau^{th}$ NN checkpoint into weighted graph $G_i^{(\tau)}$ at timestamp/epoch $\tau$. We note that although our proposed graph representation involves node features, our framework does not leverage them, and thus we consider the generated graphs unattributed. Therefore, each NN $N_{tr_i}$ is mapped to a time-evolving graph $\mathcal{G}_{tr_i} = \{\mathbf{G}_i^1, \mathbf{G}_i^2, ..., \mathbf{G}_i^t\}$. The output of this step is a set of $n$ time-evolving graphs $\{\mathcal{G}_{tr_1}, \mathcal{G}_{tr_2}, ..., \mathcal{G}_{tr_n}\}$ corresponding to the original $n$ NNs in $\mathcal{N}$.

**(S2)** *Feature extraction.* Next, the goal is to capture the structural dynamics of the NN training process. We aim to select graph measures that can capture changes during the training process, take into account the edge weight of graphs and can be calculated efficiently. In order to do that in an interpretable way, we extract two well-known node centralities from each snapshot of each generated time-evolving graph $\mathcal{G}_i$: weighted degree centrality and eigenvector centrality (§ 2.1). The weighted degree is a simple function of the learnable weight matrix $\mathbf{W}$ during the training phase of NN, therefore it gives us insights into the training dynamics at the node/neuron/filter level. The eigenvector centrality is an extension of the degree centrality, which captures the highly influential nodes, and has been successfully used in neuroscience to capture the dynamic changes of real neural networks (or connectomes) (Lohmann et al., 2010). Eigenvector centrality can be used to capture importance and connectivity of filters/neurons (i.e., the nodes in our graph representation). Also, eigenvector centrality has been used for detecting communities (Newman, 2006) or clusters (Wu et al., 2013), and thus provides structural information about the clusterability of the NN, which is complementary to that provided by the simpler and more efficient-to-compute degree centrality.

Our choice of features is also guided by the inherent $k$-partite structure of our proposed graph representation, which cannot be represented well by several other commonly-used graph features. For example, clustering-based features (e.g., number of triangles, transitivity, clustering coefficient) and cycle-based metrics which account for closed paths in a graph are always equal to 0 for $k$-partite graphs. Moreover, connected component-related features (i.e., strong/weak connectivity) do not capture the learned edge weights, which are important for modeling the training dynamics of NNs. Other features (e.g., betweenness centrality) tend to be computationally expensive, and would add significant overhead compared to early stopping methods.

**(S3)** *Graph signature construction.* In order to be able to compare NN-based graphs (with different number of nodes and edges), we summarize the structural changes in the generated time-evolving graphs at the *graph* level (rather than the *node* level, as in (S2)), and construct a statistical summary of the extracted node centralities (signature) per time-evolving graph $\mathcal{G}_i$. For each snapshot $\mathbf{G}_i^{(\tau)}$ of $\mathcal{G}_i$, we create a signature vector using five node feature aggregators, which were introduced in (Berlingerio et al., 2012) for graph similarity: *median, mean, standard deviation, skewness, and kurtosis*, where all but the median are moments of the corresponding distribution. Thus, $\mathbf{G}_i^{(\tau)}$ is mapped to a (static) signature vector $s_i^{(\tau)} \in \mathbb{R}^5$, representing the statistical summary of its node features (i.e., degree or eigenvector centrality) at time $\tau$. To put more emphasis on the most recent timestamp, we can redefine the signature at time $\tau$ as the linear weighted average of the signatures up to that point, $\mathbf{s}_i^{(\tau)} \leftarrow \frac{\sum_j^\tau j * \mathbf{s}^{(j)}}{\sum j}$, or an exponential function of the previous signatures, $\mathbf{s}_i^{(\tau)} \leftarrow \alpha \mathbf{s}_i^{(\tau)} + (1 - \alpha)\mathbf{s}_i^{(\tau-1)}$. To obtain the temporal signature of the evolving graph $\mathcal{G}_i$, we aggregate the (static) signatures up to timestamp/epoch $t$: $\mathbf{s}_i^t = \mathbf{s}_i^1 \oplus \mathbf{s}_i^2 \oplus ... \oplus \mathbf{s}_i^t$, where $\oplus$ denotes concatenation. We note that global features such as algebraic connectivity, modularity, and average shortest paths may be seen as alternative ways for constructing global graph signatures while circumventing the local feature extraction step (S2); however, these features fail to capture the structural changes in our proposed graph representations (they remain (near-)constant over time) and lead to poor performance.

**(S4)** *Performance prediction.* For the last step of performance prediction, we consider two tasks:

Table 2: Information for the generated NNs: range for early stopping epoch, range for accuracy, and accuracy threshold used for defining the class labels for classification task (predicting the accuracy level of NNs).

| | CIFAR-10 | | | | | ImageNet | | |
|---|---|---|---|---|---|---|---|---|
| | **LeNet** | **AlexNet** | **VGG** | **ResNet-32** | **ResNet-44** | **LeNet** | **AlexNet** | **ResNet-50** |
| Early stopping | $11 \sim 50$ | $30 \sim 50$ | $45 \sim 50$ | $16 \sim 120$ | $16 \sim 120$ | $16 \sim 50$ | $16 \sim 50$ | $16 \sim 120$ |
| Acc. range | $9.4 \sim 73.8$ | $5.5 \sim 82.4$ | $8.8 \sim 87.6$ | $8.4 \sim 90.0$ | $9.9 \sim 89.8$ | $0.6 \sim 14.4$ | $0.6 \sim 20.1$ | $0.86 \sim 41.66$ |
| Acc. thres. | 40 | 40 | 40 | 40 | 40 | 9 | 10 | 25 |

- Classification: We train a classifier (e.g., SVM, MLP) using the training graphs $\{\mathcal{G}_{tr_1}, \mathcal{G}_{tr_2}, ..., \mathcal{G}_{tr_n}\}$ represented by their temporal signatures $\{\mathbf{s}^t_{tr_1}, \mathbf{s}^t_{tr_2}, ..., \mathbf{s}^t_{tr_n}\}$, and their corresponding accuracies $\mathcal{A} = \{\alpha_1, \alpha_2, \ldots, \alpha_n\}$ mapped to labels $\mathcal{L} = \{l_1, l_2, \ldots, l_n\}$ (e.g., high/low accuracy) based on some threshold. Any test NN instance is then classified using the trained classifier.

- Linear regression: We perform linear regression to estimate the actual accuracy value $\alpha_{tst}$ of a new test instance $N_{tst}$ based on its signature obtained through steps (**S1**)-(**S3**).

## 4 EMPIRICAL ANALYSIS

In this section, we empirically evaluate the effectiveness and efficiency of our framework in the classification and regression tasks for different graph representations (rolled and unrolled graphs for conv layers) and different feature-based signatures (degree- vs. eigenvector centrality-based).

**Data.** We investigate NN dynamics using our framework on two well-known image classification datasets, CIFAR-10 (Krizhevsky, 2009) and ImageNet (Russakovsky et al., 2015). CIFAR-10 consists of 50K training images and 10K test images. For ImageNet, we use a sample that has 50K training images and 5K validation images used as the test set. The sample is obtained by randomly selecting 100 classes from Tiny ImageNet (tin) and downsizing the images to $32 \times 32$ colored images.

**Configuration.** The configuration of training different NN models, and the early stopping method are described in App. A.1. For the unrolled graph representation, which is signed, we consider different graph types (e.g., positive, negative), which we describe in App. A.2.

### 4.1 CLASSIFICATION: PREDICTING NN ACCURACY RANGE FROM NN TRAINING DYNAMICS

**Task setup.** We cast the NN performance prediction as a classification task. Specifically, the generated time-evolving graphs are labeled as high and low accuracy based on the performance of their corresponding NNs; Table 2 lists the threshold value chosen for low and high accuracy labels based on the final accuracy range of trained NNs, as well as the early stopping epochs for each architecture. Five-fold cross validation is used to predict the label of the test graphs in a binary classification task using SVM and MLP, where the input is the set of temporal signatures $\{\mathbf{s}^t_{tr_1}, \mathbf{s}^t_{tr_2}, ..., \mathbf{s}^t_{tr_n}\}$. We report the classification accuracy. Since the sample of NNs (App. A.1) is randomly selected with balanced high/low accuracy instances, the accuracy of a random classifier as the baseline is 50% (omitted from the charts to avoid clutter). Additionally, to show that our proposed graph representation and signatures are general and can be useful across different NN architectures, we consider the following setup: we train the classifier on a small set of NN models (i.e., different architectures—such as LeNet, AlexNet, and VGG—*and* hyperparameters), and predict the performance on *unseen* architectures (e.g., ResNet). We describe these experiments in App. A.4.

**Results.** Figure 4 illustrates the performance of SVM and MLP classifiers operating on **weighted degree**-based signatures of the rolled and unrolled graph representations of the LeNet and AlexNet architectures, trained on the CIFAR-10 dataset. We omit the results on VGG and ResNet, as well as image classification on ImageNet, because the unrolled graph generation process is prohibitively expensive, both in terms of time and space. Overall, the rolled and unrolled graph representations show similar trends in classifying NNs by effectively capturing their early training dynamics: the structural changes in the training NN architectures during the first 6-15 iterations are sufficient to classify the performance of NN instances with over 90% accuracy. However, as we discuss in § 4.3, our proposed rolled representation is significantly more efficient than the unrolled representation, and can generalize to deeper and larger NNs. We provide more details for these experiments in App. A.2.

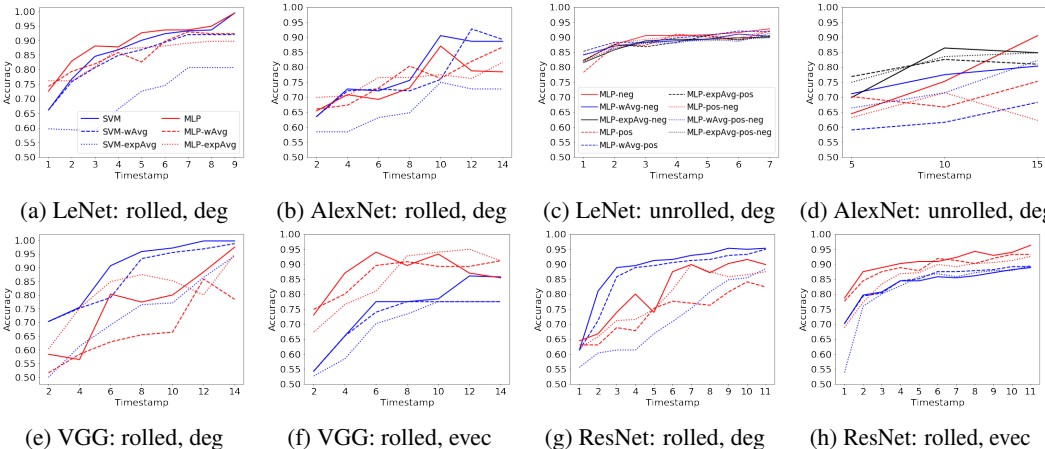

| (a) LeNet: rolled, deg | (b) AlexNet: rolled, deg | (c) LeNet: unrolled, deg | (d) AlexNet: unrolled, deg |
|---|---|---|---|
| (e) VGG: rolled, deg | (f) VGG: rolled, evec | (g) ResNet: rolled, deg | (h) ResNet: rolled, evec |

Figure 4: CIFAR-10: NN performance classification for different NN architectures, graph representations, and features for the temporal signatures. Shorthands: 'deg' for degree-based and 'evec' for eigenvector centrality-based temporal signature. **(a)–(d)**: Accuracy based on weighted degree-based signature vectors for both the rolled and unrolled graph representations. Our rolled graph representation is as effective as the unrolled representation in predicting the image classification performance of NNs, while being significantly more efficient. For the unrolled representation ((**c**) and (**d**)), the negative subgraph (solid lines) results in the most accurate performance prediction among the three subgraphs. **(e)–(h)**: Accuracy based on the weighted degree- and eigenvector centrality-based signature vectors of the rolled graph representations of VGG and ResNet-44. For both architectures, SVM performs best when leveraging the degree-based signatures (90% accuracy after 5 training epochs), while MLP outperforms SVM when operating on the eigenvector centrality-based signatures.

**OBSERVATION 1** *Both the rolled and unrolled time-evolving graph representations of NNs are effective in capturing the changes in the NN dynamics during the training phase, and can be used to predict the accuracy of an NN instance after observing only a few training epochs. Our proposed rolled representation is also space- and time-efficient, unlike the unrolled representation.*

In the remainder of this analysis, we focus on the rolled representation, which is more efficient for larger NN models and datasets. We present the classification results for both types of signatures for the temporal graphs corresponding to the training dynamics of VGG and ResNet-44 (CIFAR-10 dataset) in Fig. 4(e)-(h) and LeNet, AlexNet and ResNet-50 (ImageNet datatset) in Fig. 5. In addition to the results discussed above for LeNet and AlexNet, we provide the NN classification accuracy for the eigenvector-based signatures in Fig. 11 in the appendix. In all the cases, classification accuracy of 80-95% is achieved in less than 10 training epochs. For degree-based signatures, SVM tends to outperform MLP, while the trend is reversed for eigenvector-based signatures. For example, for both VGG and AlexNet for the CIFAR-10 image classification task, MLP can predict the performance with accuracy ∼95% using the eigenvector-based signatures from the first 6 training epochs; the same trend is observed on ImageNet for the LeNet and AlexNet architectures. For ResNet on the ImageNet classification task, SVM tends to perform well for both degree and eigenvector centrality signature vectors; while MLP tends to perform poorly in this case, the MLP variant operating on the exponential average of signature vectors outperforms the original MLP and all SVM variants. In all the cases, both classifiers reach performance over 80%-90% significantly before the early stopping point for all the architectures. In general, for ImageNet, we observe that the signatures based on eigenvector centrality are more effective in the NN performance prediction task compared to the weighted degree-based signatures—irrespective of NN architecture.

**OBSERVATION 2** *For the CIFAR-10 image classification task, the **rolled graph** representation for all NN architectures and both signature types achieve accuracy of >90%. For ImageNet, the eigenvector centrality-based signatures tend to yield higher performance compared to the weighted degree-based signatures, though both achieve accuracy of 80-90%.*

## 4.2 REGRESSION ANALYSIS: PREDICTING NN ACCURACY FROM NN TRAINING DYNAMICS

In this section, we discuss the results and key findings of predicting the actual performance (accuracy) of an NN using a linear regression model, instead of treating this task as a classification problem as in the previous section.

Table 4: Regression Analysis on **CIFAR-10**: Accuracy prediction based on the rolled graph representation and the degree-based signature.

| Time | LeNet | | AlexNet | | VGG | | ResNet | |
|------|-------|-----|---------|-----|-----|-----|--------|-----|
| | MAE | $R^2$ | MAE | $R^2$ | MAE | $R^2$ | MAE | $R^2$ |
| 0-2 | 14.95 | 0.16 | 18.712 | 0.62 | 17.72 | 0.62 | 19.84 | 0.60 |
| 1-3 | 14.67 | 0.3 | 18.57 | 0.66 | 11.98 | 0.81 | 19.41 | 0.62 |
| 3-5 | 17.09 | 0.67 | 18.4 | 0.59 | 11.91 | 0.81 | 17.63 | 0.67 |
| 5-7 | 12.49 | **0.75** | 16.74 | **0.66** | 6.87 | **0.94** | 18.28 | **0.68** |

**Task setup.** Following the same experimental methodology as in the classification task, we use the degree-based signatures of the generated time-evolving graph as the predictor and the overall accuracy of the corresponding NN as the dependent variable. We use 20% of our observations as a test set, and the rest as a training set. We report the testing mean absolute error (MAE) and coefficient of determination ($R^2$).

Table 3 displays the results for regression prediction on ImageNet dataset, the prediction model show similar trend as CIFAR-10 on this datatset. The MAE values are smaller for this dataset since the accuracy range of both architectures are much narrower than the CIFAR-10 dataset (See Table 2 for accuracy ranges of different architectures and datasets.)

**Results.** Table 4 summarizes the regression results for four architectures where the independent variable is degree features of rolled graph representation. For each step of this experiment, we use a concatenation of two consecutive feature vectors as independent variable of regression model. For LeNet, at timestamp 5-7 the regression model error (MAE for test set) is 5.74, while the accuracy range of LeNet for this dataset is wide (0.9-73.8), and based on $R^2 = 0.91$, we can interpret that 91% of variant in accuracy of NN can be explained by the degree changes of the rolled graph between timestamp 5-7. A similar trend is observed for AlexNet, VGG and ResNet, where after observing a few epochs, the prediction model shows a very low MAE with high $R^2$.

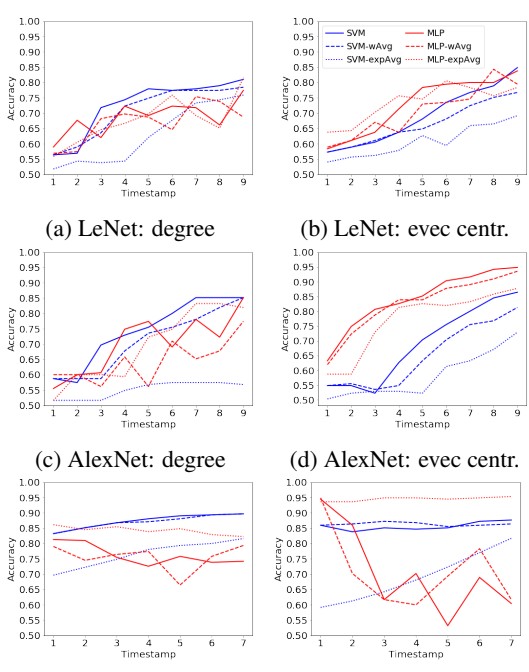

(a) LeNet: degree

(b) LeNet: evec centr.

(c) AlexNet: degree

(d) AlexNet: evec centr.

(e) ResNet-50: degree

(f) ResNet-50: evec centr.

Figure 5: ImageNet: NN classification using the weighted degree-based and eigenvector centrality-based signature vectors for LeNet, AlexNet and ResNet. Eigenvector centrality-based signatures (**b**), (**d**), (**f**) yield higher performance compared to the weighted degree-based signatures (**a**), (**c**), (**e**).

**OBSERVATION 3** *In sum, the weighted degree-based signature vector of time-evolving graphs generated based on the rolled graph representation is a strong predictor of the actual accuracy value of NNs. For both CIFAR-10 and ImageNet, we show that, by observing only a subset of early training epochs, we can effectively predict the accuracy value of NNs with a small $MAE$ and a high coefficient of determination $R^2 > 0.5$.*

## 4.3 DISCUSSION

**Time efficiency and early stopping.** Figure 6 represents the total average runtime of NN training for early stopping of each architecture (green bars) with comparison of average run-time of graph generation, degree calculation and eigenvector centrality calculation for the number of epochs that were needed for that architecture to achieve the highest accuracy in classification task. For all the 3 architectures

Table 3: Regression Analysis on **ImageNet**: Accuracy prediction based on the rolled graph representation and the degree-based signature.

| Time | AlexNet | | LeNet | | ResNet | |
|------|---------|-----|-------|-----|--------|-----|
| | MAE | $R^2$ | MAE | $R^2$ | MAE | $R^2$ |
| 0-3 | – | – | 4.04 | 0.30 | 7.99 | 0.57 |
| 3-5 | 4.10 | 0.65 | 3.94 | 0.34 | 8.36 | 0.60 |
| 5-7 | 3.96 | 0.70 | 3.58 | 0.46 | 8.02 | 0.62 |
| 7-9 | 2.70 | **0.84** | 3.24 | **0.54** | 7.49 | **0.67** |

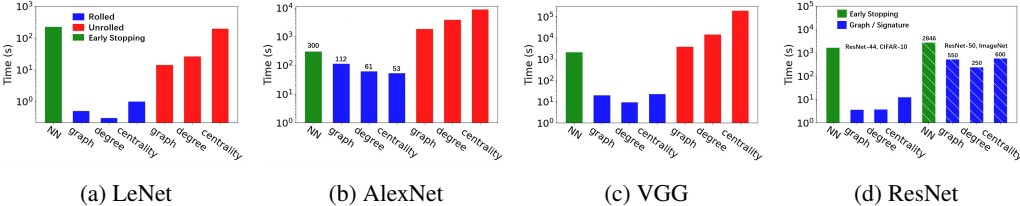

Figure 6: Average total runtime for the NN training, and the rolled/unrolled graph generation (**S1**) and feature extraction (**S2**) on CIFAR-10 dataset ((**a**)-(**c**)), ResNet architecture on both CIFAR-10 (ResNet-44) and ImageNet (ResNet-50)(**d**). Rolled graph representation is more efficient/faster than early stopping to generate graph and calculate signature vector for all the architectures.

rolled graph representation can achieve a high accuracy prediction much faster than the early stopping of NN training. But for unrolled representation, this is only true for LeNet architecture. As the size of NN increases, the unrolled graph generation gets slower and training with early stopping method is faster than our prediction framework.

**Size analysis.** Table 5 shows the number of nodes and edges of the rolled graph representations compared with the unrolled representation, our proposed rolled representation of NN reduces the complexity of NN architectures in terms of #nodes and #edges in the graph while retaining comparable performance on the prediction tasks. Our proposed rolled graph representation has big advantage in terms of size over the unrolled graph representation, especially for deeper networks.

**Limitations.** The unrolled representation is not scalable therefore cannot be applicable for larger networks and datasets. The main advantage of this representation is that there is no loss of information during the training phase of NNs, and we can interpret the learning process easier than the rolled method. For example, we showed that the negative subgraph alone would be a stronger predictor of the accuracy of NNs, which can lead us to another direction of signed graph analysis in further study of the learning process of NNs.

**Complexity analysis of the rolled graph representation.** For an NN with $n_f$ total number of filters in all the conv layers and $n_n$ total number of neurons in the fully connected layers, modeling the nodes takes a constant time $O(|\mathcal{V}|)$, where $\mathcal{V}$ is the set of generated nodes in the graph and $|\mathcal{V}| <= n_f + n_n$ (due to dropout some nodes are removed). The computation of the edge weights is $O(\sum_l |\mathcal{V}|_l * |\mathcal{V}|_{l+1})$, where $|\mathcal{V}|_l$ are the nodes at layer $l$.

## 5 RELATED WORK

We cover the most relevant work here, and dynamic graph mining in App. A.5. Some recent research efforts are devoted to modeling the NN architectures as graphs due to their topological identity and study their graph properties Rieck et al. (2019); Filan et al. (2020); You et al. (2020). For example, You et al. You et al. (2020) propose a relational graph representation to model message exchange between layers, and empirically show the common properties shared by NNs with significantly improved predictive performance in terms of graph clustering coefficients and average path lengths. Rieck et al. Rieck et al. (2019) propose a complexity measure related to NN performance—neural persistence—based on topological data analysis on weighted stratified graph. Filan et al. Filan et al. (2020) present an exploratory study of the NN modularity. Gebhart et al. (2019) proposes to compute persistent homology over the activation graph of an NN. The output is a graded set of subgraphs, which are shown to be related to the task-specific semantic that are captured by original NN.

## 6 CONCLUSION

In this work, we investigated the early training dynamics of NNs from a time-evolving graph perspective. We proposed a new graph representation to efficiently convert convolutional layers into compact and intuitive graph structures. Then, we showed that a simple, temporal graph signature based on summary statistics of the degree or eigenvector centrality distributions over *only a few epochs* can be used as a strong predictor variable to estimate the accuracy of NNs in downstream tasks (e.g., image classification). Exploring the role of our efficient proposed framework for early stopping is a promising future direction.

## 7 REPRODUCIBILITY STATEMENT

For reproducibility, we provide the references to the datasets and existing (rolled) graph representation of convolutional layers in § 4. In App. § A.1, we provide the detailed hyperparameter settings for NN training. We will also make our code publicly available upon acceptance.

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

# A APPENDIX

## A.1 EXPERIMENTAL SETUP: CONFIGURATION

We first train five different NN architectures, LeNet-5 (Lecun et al., 1998), VGG13 (Simonyan & Zisserman, 2015), AlexNet (Krizhevsky et al., 2012), ResNet-32 and ResNet-44 (He et al., 2016) , on CIFAR-10 dataset and three architectures, LeNet-5, AlexNet and ResNet-50, on ImageNet. During training, the NN parameters are updated using stochastic gradient descent. On each dataset, we train NNs by combining 48 learning rates $\{1, 1.5, \ldots, 4.5\} \times \{10^{-6}, \ldots, 10^{-1}\}$, and 10 dropout rates $\{0, 0.1, \ldots, 0.9\}$. All the models are trained for 50 epochs with batch size 128 and an **early stopping method** by which the training stops when the testing accuracy does not increase for 10 consecutive epochs. Consequently, we obtain 480 NNs per architecture and dataset, with diverse performance. Information for the generated NNs is summarized in Table 2: the range of epochs at which the training stops early, the range of final testing accuracy for the trained NNs, and the accuracy threshold used to map actual NN performance to 'low/high accuracy' labels for the classification task. We trained the NNs on an Nvidia 1080Ti GPU with 11G memory, and we conducted all the other experiments on 2.60GHz Intel Xeon E5-2697 v3 platform with 1024G memory.

For computational efficiency and to avoid having a largely imbalanced dataset, out of the 480 NNs, we randomly sampled 250 NNs with an even split of high- and low-accuracy networks. For this sample, the first $t$ epochs of training for each configuration were saved as checkpoints to be converted to time-evolving graphs (step **(S1)**).

## A.2 UNROLLED GRAPH REPRESENTATION

In these experiments, we split the represented graph into two subgraphs with positive and negative edge weight [1]. Then we used the two graphs $\mathbf{G}_{pos}$ and $\mathbf{G}_{neg}$ to calculate the degree and eigenvector centrality feature vector summary for both of them, and concatenate those feature vectors to represent the whole graph $\mathbf{G}_{pos-neg}$. According to Figure 4, **weighted degree signature vector** of $\mathbf{G}_{neg}$ is a good descriptive variable to predict performance of NNs. For LeNet, we achieve an accuracy of 90% with only 3-5 epochs. For AlexNet, since the graph generation process was slow, we constructed the time-evolving graph based on a few sub-samples of training epochs where $T = 1, 5, 10, 15$. The results show that even with a fewer sequence of observations the classifier can achieve a high value of accuracy. For instance, at timestamp 10, in which the feature vector is the concatenation of degree features at epochs 1, 5, and 10, the accuracy of the classifier is more than 85%.

Based on results in Figure 11a, signature vector calculated based on **eigenvector centrality** of $\mathbf{G}_{neg}$ is also very good descriptive variable to predict performance of NNs and can achieve accuracy of 90% with only 3-5 epochs for LeNet.

For all the experiments on unrolled representation, the feature vector only based on $\mathbf{G}_{neg}$ achieves very high accuracy similar to or higher than that of $\mathbf{G}_{pos-neg}$, while feature vectors only based on $\mathbf{G}_{pos}$ result in the poorest accuracy among them.

## A.3 FEATURE ANALYSIS

**Classification feature analysis.** Figure 7a shows the weight of each statistical aggregator for graph degree of VGG architecture for CIFAR-10. The top three highly weighted features are mean, standard deviation and median of degree. Figures 7b, 7c and 7d represent the change in value of those features respectively over time for high and low accuracy instances. We observe that the statistical aggregator of degree for high accuracy graphs tends to increase over time while for most of low accuracy one these values do not change over time or the change is very small. Figure 8 depicts similar trend for the LeNet architecture, while the values of degree mean and standard deviation for low accuracy cases tend to have a very big spike in the first 3 epochs of training but after that they follow a flat line trend and tend to not change. Figure 9 illustrates similar trend for ResNet-44 architecture and CIFAR-10 dataset. The mode degree of graphs based on high accuracy cases tends to change drastically over time while the low accuracy cases do not change and show a flat line pattern.

---

[1] In the rolled graph representation there is no negative weighted edge since the weight of each edge is calculated as a norm of each channel

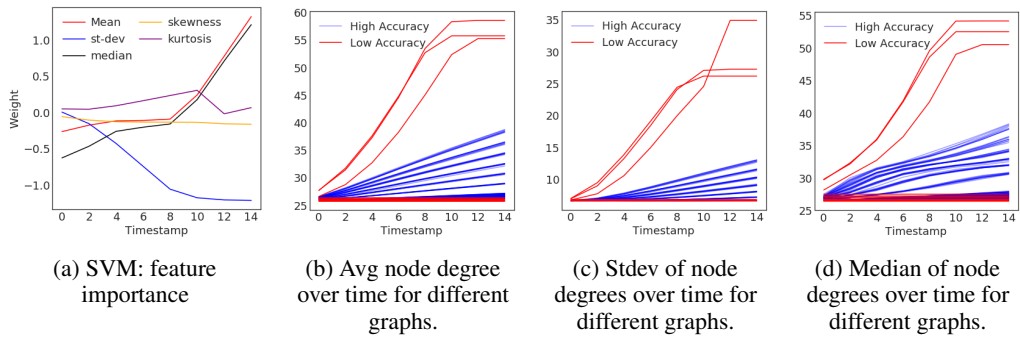

(a) SVM: feature importance

(b) Avg node degree over time for different graphs.

(c) Stdev of node degrees over time for different graphs.

(d) Median of node degrees over time for different graphs.

Figure 7: VGG on CIFAR-10: **(a)** The top three most important features in SVM classification are the mean, stdev, and median of the node degrees. **(b)** The change of the average node degree over time shows that, in graphs corresponding to high-accuracy NNs (blue lines), the average degree exhibits an increasing pattern, while it has a flat pattern for low-accuracy cases (red lines). The few cases of low-accuracy NNs with increasing trend may be miss-classified in the classification task. **(c) (d)** The changes of the standard deviation and median of node degrees follow a similar pattern to mean for both low- and high-accuracy cases.

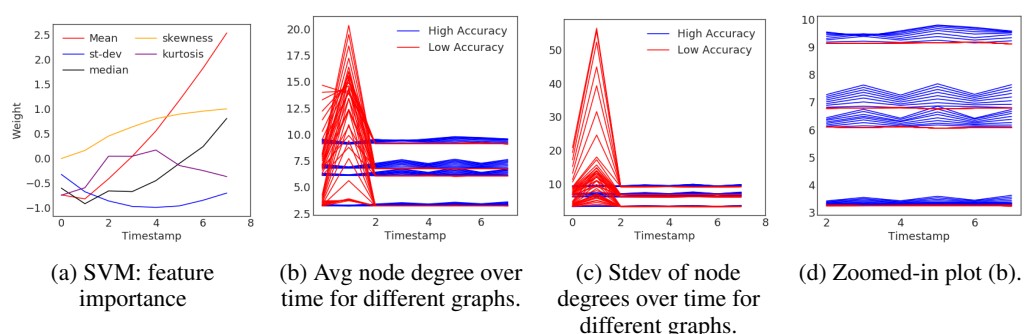

(a) SVM: feature importance

(b) Avg node degree over time for different graphs.

(c) Stdev of node degrees over time for different graphs.

(d) Zoomed-in plot (b).

Figure 8: LeNet on CIFAR-10: **(a)** The top three most important features in SVM classification are the mean, standard deviation and median of the node degrees. **(b)(d)** The change of average node degree over time shows that in graphs corresponding to high-accuracy NNs (blue lines) the average degree tends to change over time with a smooth pattern, while it tends to follow a flat line pattern (after an early extreme spike) for low-accuracy cases (red lines).

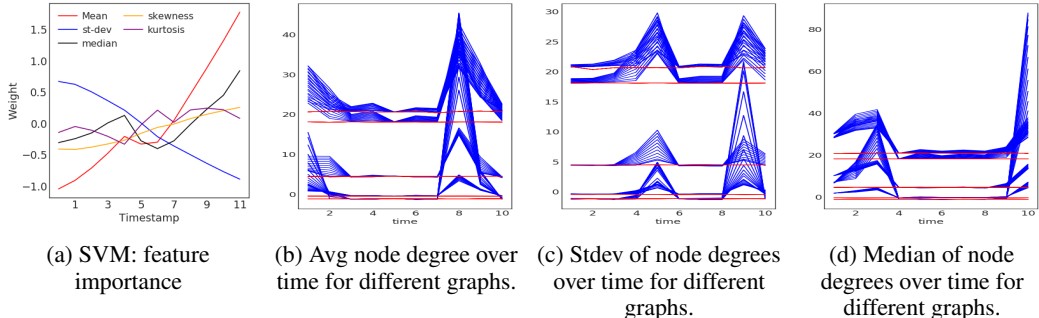

(a) SVM: feature importance

(b) Avg node degree over time for different graphs.

(c) Stdev of node degrees over time for different graphs.

(d) Median of node degrees over time for different graphs.

Figure 9: ReNet-44 on CIFAR-10: **(a)** The top three most important features in SVM classification are the mean, standard deviation and median of the node degrees. **(b)(d)** The change of average node degree over time shows that in graphs corresponding to high-accuracy NNs (blue lines) the average degree tends to change over time drastically, while it tends to follow a flat line pattern for low-accuracy cases (red lines).

## A.4 GENERALIZING TO UNSEEN ARCHITECTURES

In this section, we broaden the scope of our empirical setup to show that our proposed graph representation and signatures are general and can be useful for performance prediction across different NN architectures. In this setup, we fully train just a small set of different NN architectures (i.e., different architectures and hyperparameters), and predict the performance on unseen NN architectures. We consider two sets of experiments:

(a) CIFAR-10, ResNet-44   (b) ImageNet, ResNet-50   (c) CIFAR-10, ResNet-32   (d) CIFAR-10, ResNet-44

Figure 10: NN classification based on **degree** signatures for the empirical setup that tests generalization to unseen architectures. **(a)** Train on ResNet-32, test on ResNet-44. **(b)** Train on ResNet-34, test on ResNet-50. **(c)-(d)** Train on LeNet, AlexNet and VGG, test on ReNet-32 and ResNet-44, respectively. Our proposed framework is able to accurately predict the performance level on previously unseen architectures based on the NN structural dynamics in a small number of epochs (<10).

1. We train our classifier on the smaller ResNet architectures (ResNet-32 for CIFAR-10, and ResNet-34 for ImageNet), and test the performance of the larger ResNet architectures (ResNet-44 for CIFAR-10, and ResNet-50 for ImageNet). In Figs. 10a and 10b, we observe that our proposed framework is able to accurately predict the performance level of the previously unseen, large ResNet architectures. Our results show that the proposed temporal signatures can be used in a generalized scenario to predict the accuracy level of the same architecture with different numbers of layers (on the same dataset). This generalization from small to bigger architectures is important since it is faster to train the smaller architectures.

2. In Figs. 10c and 10d, we see that our proposed method also successfully predicts the performance level of a new architecture (i.e. ResNet) when the training set is a combination of older architectures (LeNet, VGG, AlexNet).

For both of the experiments, we set a universal threshold value to label graphs in the train and test set. For the training set of classifiers, we randomly choose a subset of the NNs with balanced high/low accuracy labels. The size of the training set in the first experiment is 250 and in the second experiment is 400. The experiments were repeated 5 times and the average accuracy of classification is reported.

## A.5 ADDITIONAL RELATED WORK

**Dynamic Graph Mining.** Dynamic graphs are mostly modeled as a sequence of edge additions and/or edge deletions. To mine the dynamic graphs, traditional approaches leverage graph properties such as node centralities or motifs Riondato et al. (2017); Paranjape et al. (2017); Kovanen et al. (2011). Despite their simplicity, these approaches show effectiveness in temporal tasks such as event detection Paranjape et al. (2017) and structural prediction Aggarwal & Subbian (2014). Recent embedding-based approaches tend to model dynamic graphs as a sequence of discrete-time snapshots and simultaneously represent the graph structure of each snapshot as well as the temporal evolution using deep neural networks such as GRU/LSTM Sankar et al. (2020); Singer et al. (2019); Pareja et al. (2020). DySAT Sankar et al. (2020) leverages self-attention to compute node representations by jointly modeling graph structural property and temporal dynamics. EvolveGCN Pareja et al. (2020) uses GCN to generate node embeddings for the past snapshots, and learns parameters of the hidden layer for the next snapshot using GRU/LSTM. Unlike these methods, we propose two different approaches to represent the underlying graph structure of NN, and use a very simple and efficient dynamic graph signature feature vector to predict the accuracy of corresponding NNs.

### A.6 Additional Figures and Tables

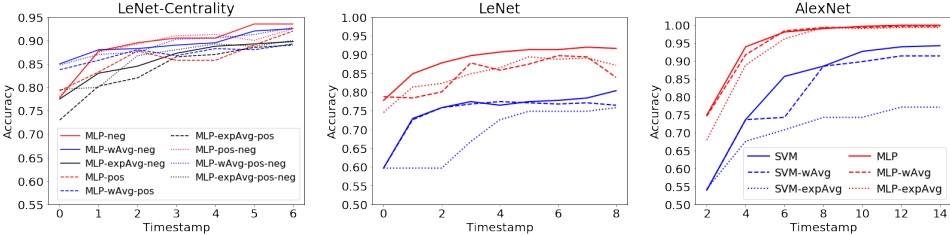

(a) LeNet, unrolled graph repre-
sentation

(b) LeNet, rolled graph represen-
tation

(c) AlexNet, rolled graph repre-
sentation

Figure 11: CIFAR-10: NN classification based on **eigenvector centrality-based signatures**. The eigenvector centrality is a strong predictor of NN performance after observing a few epochs of training. The MLP classifier (red lines) outperforms SVM for all the architectures.

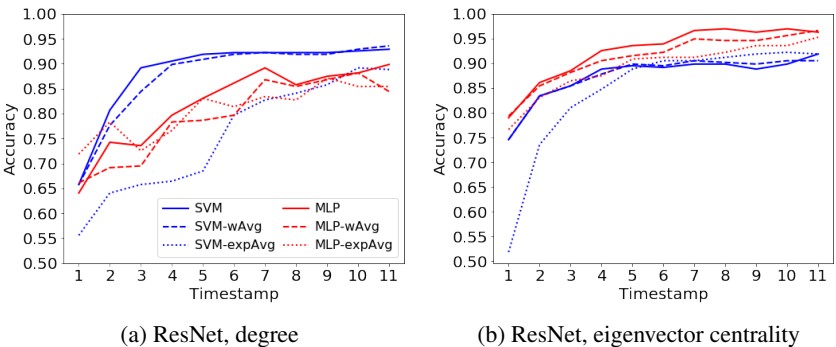

(a) ResNet, degree

(b) ResNet, eigenvector centrality

Figure 12: CIFAR-10, ResNet-32: NN classification based on **degree** and **eigenvector centrality-based** signatures. The both degree and eigenvector centrality are strong predictor of NN performance after observing a few epochs of training.

Table 5: Size of a generated graph snapshot based on the rolled and unrolled conv layer representation.

|  | CIFAR-10 | | | | | ImageNet | | |
|---|---|---|---|---|---|---|---|---|
|  | **LeNet** | **AlexNet** | **VGG** | **ResNet32** | **ResNet44** | **LeNet** | **AlexNet** | **ResNet50** |
| $|\mathcal{V}|$, rolled | 239 | 2 925 | 1 613 | 1 149 | 1 597 | 713 | 3 015 | 22 823 |
| $|\mathcal{E}|$, rolled | 12 954 | 1 160 480 | 304 576 | 51 888 | 73 392 | 115 362 | 1 206 560 | 10 821 824 |
| $|\mathcal{V}|$, unrolled | 11 166 | 54 986 | 205 066 | | | 11 640 | 55 076 | |
| $|\mathcal{E}|$, unrolled | 658 024 | 45 997 696 | 119 714 496 | | | 843 376 | 46 043 776 | |

