# OpenReview forum: "Convolutional Neural Network Dynamics: A Graph Perspective"
_ICLR.cc/2022/Conference — ICLR 2022 Submitted_

### Official Review · Reviewer_DNSD · 2021-10-30

**Correctness:** 2
**Technical Novelty And Significance:** 3
**Empirical Novelty And Significance:** 2
**Recommendation:** 3
**Confidence:** 5

**Main Review:**

Strength: It appears as though the rolling graph representation is promising. The subject is novel and deserves more investigation.

Weakness:
 - It's unclear why only degree and Eigen centrality are taken into account. There are other alternative measures of importance or centrality.
- The proposed method is intuitive and lacks solid theoretical support.
- The overview of node characteristics does not adequately reflect the structural evolution of large-scale NNs in the modern era (ResNet).
- The neural networks empirically evaluated, including LeNet, AlexNet, and VGG, are somewhat out of date. It would be advantageous to consider some more recent NNs, such as ResNet.
- To facilitate interpretation, node centralities are used to indicate the graph characteristic. Reintroducing them into SVM may not improve their interpretability in some way.
- To make the results more understandable, the centralities can also be put into a more interpretable classification model, such as logistic regression or classification tree.
- Baselines are omitted from the experiment.

**Summary Of The Paper:**

The purpose of this research is to attempt to forecast a neural network's performance from early NN dynamics. The dynamics of the neural network are stored in a time-evolving graph. The summary of two node properties, the weighted degree, and the eigenvector centrality is extracted and utilized to train classification and regression algorithms. This methodology, when combined with the paper's suggested rolled graph representation for convolutional layers, results in faster prediction than the classic early stopping method.
The paper's primary contribution is that the suggested rolled graph model for convolutional layers improves time and space efficiency in comparison to the existing unrolled representation.

**Summary Of The Review:**

I recommend rejecting this article. Although predicting NN performance from the network structure and dynamics is an intriguing topic, it is not one that this research proposes. The method, as well as the experiment, are more about preliminary and experimental exploration. Apart from the rolled graph representation of the convolutional layers, there is only a little theoretical enhancement.

---

> ### Author Response · Authors · 2021-11-23
> **Response to reviewer  DNSD part 1**
>
> Thank you for your time and feedback.
>
> 1- “Why degree and eigenvector centrality”
> Based on your feedback, we have updated our paper (Sec. 3.2 (S2) and (S3)) with additional justifications and intuition behind the graph features that we leverage during the signature construction step (degree and eigenvector centrality).
> We chose weighted degree centrality to efficiently capture (1) the _weights_ that are learned in each layer of the NN at the node level (filter or neuron), which are the most important aspect of the training process;  and (2) the _change_ of the learned weights during the training process of NN over time (in the proposed time-evolving graph representation). Moreover, we use the node degrees to model the NN structure since they have been found to be effective in capturing the structural roles of nodes and a key notion for structural node embedding [1]. We note that we show the difference in how the degree-based signatures change during the training process for low accuracy NNs vs. high accuracy ones in Figure-7-9 in the appendix.
> Eigenvector centrality is used to capture the highly influential nodes in a graph; in our case, the nodes correspond to filters of convolutional layers and neurons of fully connected layers. Additionally, eigenvector centrality has been used for detecting communities [2] or clusters [3], and thus provides structural information about the clusterability of the NN. Clusters or communities are central for network analysis and provide complementary information compared to the simpler and more efficient-to-compute degree centrality. Our hypothesis was that tracking changes in the eigenvector centrality (or node influence or clustering-related information) over time could provide complementary insights into the structural changes of the NN during training that may correlate with high or low performance.
>
> *Global vs. local properties*: Our proposed framework requires global graph summaries in order to be able to compare different NN-based graphs (with different number of nodes and edges) in the training set; we achieved this by computing statistical summaries (signatures) of graph properties that capture the structure locally at the node level and are efficient to compute (so that our approach adds little overhead). We also experimented with other global features such as algebraic connectivity, modularity, and average shortest paths, but those fail to capture the structural changes in our graphs and lead to poor classification performance (they remain constant at different timestamps of the represented graphs).
>
> *Why not use other graph properties?* Our proposed graph representation is k-partite, and thus many commonly used graph features are not appropriate.  For example, clustering-based features (such as number of triangles, transitivity, clustering coefficient, square clustering coefficient), and cycle-based metrics which account for closed paths in a graph (e.g., cycle basis, minimum cycle basis) are always equal to 0 due to the k-partite structure of our graphs, and so fail to model the graph structure. Connected component-related metrics (i.e., connectivity, strong connectivity, weak connectivity) do not capture the weighted edges (i.e., the learned weights), which are important in our scenario for modeling the training dynamics of NNs. In terms of other centralities, betweenness centrality (which is commonly used in network analysis) is based on all the shortest paths that pass through a particular node, which makes it computationally expensive and less efficient than eigenvector centrality.
> [1] Jin, J. et al. “Toward Understanding and Evaluating Structural Node Embeddings”. ACM Transactions on Knowledge Discovery from Data 2022. arXiv:2101.05730v1
> [2] Newman, Mark EJ. "Finding community structure in networks using the eigenvectors of matrices." Physical review E 74.3 (2006): 036104.
> [3] Wu, Qin, et al. "“follow the leader”: A centrality guided clustering and its application to social network analysis." The Scientific World Journal 2013 (2013).

---

> > ### Author Response · Authors · 2021-11-23
> > **Response to reviewer DNSD part 2**
> >
> > 2- “Intuitive approach / theoretical support”
> >
> > We are glad that you find our method intuitive. Our key contribution is that we introduced a new graph representation that can model a variety of popular NNs and their learning dynamics during the training process in an intuitive way. Unlike prior work that focuses on static representations or other abstractions, our representation is time-evolving and its components (nodes, edges), which capture the learned parameters (by design), are easy-to-understand. To show the effectiveness of our graph representation, we formulated the performance prediction task, which investigates the relation between the structural changes in our graph-based NN representations and the NN outcomes (performance) for different hyperparameter configurations; we predict both the actual value of accuracy for an NN model and its performance level (high/low accuracy). We believe that our extensive empirical results are promising and highlight the value of our proposed graph model, which could open up new research directions towards understanding the NN structure and its interplay with other factors.
> >
> > To strengthen our paper and highlight the effectiveness of our model more, we also broadened the scope of our empirical setup to show that our proposed graph representation and signatures are general and can be useful _across_ different NN architectures. This allows us to fully train just a small set of different NN architectures (i.e., different architectures and hyperparameters), and predict the performance even on unseen NN architectures. Thus, we updated our paper (Appendix A.4) and added a new set of results:
> >
> > * (1) We train our classification model on the smaller ResNet architectures  (ResNet-32 for CIFAR-10, and ResNet-34 for ImageNet), and test the performance of the larger ResNet architectures (ResNet-44 for CIFAR-10, and ResNet-50 for ImageNet). In Figures 10(a)-(b), we observe that our proposed framework is able to accurately predict the performance level on the previously unseen large ResNet architectures. Our results show that the proposed temporal signatures can be used in a generalized scenario to predict the accuracy level of the same architecture with different numbers of layers (on the same dataset). This generalization from small to bigger architectures is important since it is faster to train the smaller architectures.
> >
> > * (2) We show that our proposed method also successfully predicts the performance level of a new architecture (i.e. ResNet) when the training set is a combination of older architectures (LeNet, VGG, AlexNet). We provide more details on this experiment in Appendix A.4.
> >
> >
> > 3,4- “It would be advantageous to consider some more recent NNs, such as ResNet.”
> >
> > As we have shown in our empirical analysis, our proposed method is effective in capturing the training dynamics of ResNet (in addition to other architectures).
> > We have given results on the CIFAR-10 dataset for the ResNet-32 and ResNet-44 architectures: Fig 4(g)-(h), Fig12(a)-(b) for classification, and the last column of Table 4 for regression analysis. Moreover, for ImageNet, we presented results for the ResNet-50 architecture: Fig 5(e)-(f) for classification, and the last column of Table 3 for regression.
> >
> > As we mentioned in the previous point, we have also revised our paper to include results for a less restrictive empirical setup where the test set consists of architectures that are not encountered at all in the training set. Our new experiments also include results on ResNet (Appendix A.4).

---

> > > ### Author Response · Authors · 2021-11-23
> > > **Response to reviewer DNSD part 3**
> > >
> > > 5,6- “Logistic regression for interpretability”
> > >
> > > We agree that depending on the task and objective, choosing the appropriate classifier is important. We also experimented with logistic regression, and the trend is similar to SVM. Please check Fig 10 for additional results based on logistic regression.
> > >
> > > In this work, we aim to show the effectiveness of the rolled graph representation to predict the accuracy of NNs without seeking interpretability for the predictions. We have updated our paper to phrase our statements regarding interpretability more clearly (Section 3.1 and Section 5); we realize that our previous claims could be confused with “NN interpretability”, a topic that has a different focus than our work.
> > >
> > > Our main claim is that our proposed rolled graph representation is more interpretable than the unrolled representation---i.e., it is easier to understand what it represents. In our proposed rolled graph representation, the nodes are defined to represent filters in convolutional layers; on the other hand, the nodes represent feature maps in the unrolled representation introduced by (Rieck et al, 2019). Thus, we argue that it is easier to interpret the outputs of downstream graph analysis on our graph representation (e.g., computing node properties, tracking the evolution of the graph, performing community detection). For example, when we calculate the eigenvector centrality, it is clear which filters (from a specific layer) are the most influential filters; if we were to do the same calculation on the previously introduced unrolled representation, it would not be possible to draw such a conclusion. Furthemore, we believe that our graph representation can be a powerful tool for future studies that aim to understand the NN dynamics.
> > >
> > > 7- “Baselines”
> > > To the best of our knowledge, there is no other method that aims to solve the problem that we introduced in this paper, i.e., the prediction of the accuracy of NNs. Nevertheless, we reported results on a previously proposed graph representation for convolutional layers (i.e., the unrolled  graph representation introduced by Rieck et al. 2019) in order to compare the efficiency and accuracy of our proposed rolled representation. We reported results on this “baseline” representation in Fig 4 (c) and (d). Additionally, we considered different options for other steps of our proposed methodology beyond the graph construction (e.g., different signature constructions, classification vs. regression) and discussed our observations. Finally, we compared the efficiency of our approach to the “early stopping method”.

---

> > > > ### Comment · Reviewer_DNSD · 2021-11-26
> > > > **discussion**
> > > >
> > > > The revision addressed the majority of my concerns. They justified the measures, extended the empirical experiment to ResNet, and omitted the assertion about interpretability. Also, this direction looks promissing.
> > > >
> > > > However, as you noted in your initial review, theoretical support still remains missing.

---

> > > > > ### Author Response · Authors · 2021-11-29
> > > > > **Discussion response to Reviewer DNSD**
> > > > >
> > > > > Thank you for taking the time to read our response. We are glad that you find this direction promising and that our response and updates to our paper addressed your concerns (including new experiments that complemented our experiments on ResNet in our original submission).

---

> > > > > ### Author Response · Authors · 2021-11-30
> > > > > **Discussion response**
> > > > >
> > > > > Given that our paper updates and response adequately addressed the vast majority of your concerns, we are wondering if you would be willing to increase your score? Thank you again for your time and feedback.

---

### Official Review · Reviewer_zP52 · 2021-11-02

**Correctness:** 3
**Technical Novelty And Significance:** 3
**Empirical Novelty And Significance:** 3
**Recommendation:** 5
**Confidence:** 3

**Main Review:**

Technically the paper is sound and I enjoyed reading it. However, I have some concerns regarding the presentation and motivation of the problem considered.
- The motivation for the problem is not clear to me. Sure, with this technique one could decide within a few epochs of starting training of a model, whether to continue or terminate training. However, to train such a predictor requires several rounds of training for the original task (under different hyperparameters) to already have been done. Why does one need to retrain for the original task is unclear to me.
- One of the key claimed contributions of the paper is the rolled graph representation. However, it is not clear what is the difference between a rolled representation and an unrolled representation (i.e., section 3.1). E.g., the edge between v^i_k and v^j_l has weight norm(K(l,h_j,w_i,k)) but it’s unclear what the latter notation means. It would greatly help if the authors could explain (and justify) the differences better.
- It seems bias parameters are not involved while summarizing the neural network state. Why is this the case.
- Problem 1 in Section 3 needs clarification. The paper seeks to predict the accuracy of a new instance N_tst, but the accuracy of N_tst after how many rounds of training is not specified.


**Summary Of The Paper:**

The paper considers the problem of predicting test performance of a neural network model by using statistics about the parameter weights in relation to the network structure during the first few epochs of training. The authors propose two approaches, using degree centrality and eigenvector centrality, to compute a “signature” that summarizes the neural network’s state during a training epoch. They also propose a rolled-graph representation that uses fewer nodes to represent a convolutional neural network, thereby making the training and prediction process more efficient. Experimental evaluation shows proposed method 	having similar predictive performance as unrolled representation, while being computationally faster.

**Summary Of The Review:**

Technically sounds paper, but lacking in motivation and presentation.

---

> ### Author Response · Authors · 2021-11-23
> **Response to reviewer  zP52 part 1**
>
> Thank you for your time and feedback. We are glad that you enjoyed reading our paper.
>
> 1- “Motivation”
> This problem is inspired by the growing interest in explainable AI and the need to better understand the widely-used artificial neural networks. Towards this end, we introduced a new graph representation that can model a variety of popular NNs and their learning dynamics during the training process in an intuitive way. Unlike prior work that focuses on static representations or other abstractions, our representation is time-evolving and its components (nodes, edges), which capture the learned parameters (by design), are easy-to-understand. To show the effectiveness of our graph representation, we formulated the performance prediction task, which investigates the relation between the structural changes in our graph-based NN representations and the NN outcomes (performance) for different hyperparameter configurations; we predict both the actual value of accuracy for an NN model and its performance level (high/low accuracy). We believe that the empirical results are promising and highlight the value of our graph model, which could open up new research directions towards understanding the NN structure and its interplay with other factors.
>
> As you pointed out, our methodology can decide within a few epochs whether to continue or terminate training. Moreover, our approach is faster than the early stopping method, so it can help with hyperparameter search with a _larger_ set of hyperparameters.
>
> That said, we agree with you that our setup is somewhat limited since we need some fully trained configurations in order to make predictions for a new hyperparameter configuration of the same architecture and on the same dataset. Inspired by your comment, we broadened the scope of our empirical setup to show that our proposed graph representation and signatures are general and can be useful for performance prediction _across_ different NN architectures. This allows us to fully train just a small set of different NN architectures (i.e., different architectures and hyperparameters), and predict the performance even on unseen NN architectures.
>
> Thus  we updated our paper ((Appendix A.4) and added a new set of results:
>
> * (1) We train our classification model on the smaller ResNet architectures  (ResNet-32 for CIFAR-10, and ResNet-34 for ImageNet), and test the performance of the larger ResNet architectures (ResNet-44 for CIFAR-10, and ResNet-50 for ImageNet). In Figures 10(a)-(b), we observe that our proposed framework is able to accurately predict the performance level on the previously unseen large ResNet architectures. Our results show that the proposed temporal signatures can be used in a generalized scenario to predict the accuracy level of the same architecture with different numbers of layers (on the same dataset). This generalization from small to bigger architectures is important since it is faster to train the smaller architectures.
>
> * (2) We show that our proposed method also successfully predicts the performance level of a new architecture (i.e. ResNet) when the training set is a combination of older architectures (LeNet, VGG, AlexNet). We provide more details on this experiment in Section (Appendix A.4).
>
>
> 2- “Rolled vs. unrolled representation”
> Thank you for your feedback. We updated the paper (Section 3.1) to make the definition of edge weights more clear and emphasize the differences between rolled and unrolled representations. We address your question about the main difference between the proposed graph representation and unrolled representation below.
>
> The two representations model convolutional layers in significantly different ways.
> *Nodes*: In our rolled graph representation, a node corresponds to a filter in the Conv layer; in the unrolled representation, each node is defined as a feature map that is learned through the convolutional process, which makes the number of nodes grow very fast (We describe the number of nodes and edges for the unrolled representation in Section 2.3, page 3).
> *Edges*: In our rolled graph representation, edges connect filters in consecutive layers (e.g., layer l and l+1), and they are weighted by the norm of channels in the (l+1)-layer filters. For example, the edge weight between node i_l (i.e., filter i in layer l) and j_l+1 (i.e., filter j in layer l+1) is calculated as the norm of the i-th channel of filter j. On the other hand, in the unrolled representation, the weight is the weights learned in each feature map of the convolutional operation.
> To summarize, our rolled graph is represented at the filter level, while the previously proposed unrolled representation---as the name suggests---unrolls the convolutional operation at the feature map level and represents the learned feature maps.

---

> > ### Author Response · Authors · 2021-11-23
> > **Response to reviewer zP52 part 2**
> >
> > 3- “Bias parameters”
> > In Section 3.1, we proposed to capture the biases as node features in our rolled graph representation. However, our methodology did not make use of the node features since the graph structural measures that we chose model only the graph structure. As we showed in our empirical analysis, the structural changes during the training process are very effective in predicting the NN performance. We believe that an interesting future direction is developing an efficient graph representation learning method for time-evolving k-partite graphs which captures both the structure (graph) and node features (biases)---to the best of our knowledge, such a method has not been proposed yet for this graph type.
> >
> >
> > 4- “Clarification of Problem 1”
> > Thank you for pointing this out; we updated the paper to clarify the problem statement (Section 3, Problem 1).
> >
> > When we predict the accuracy at time t for the test set, we use the exact same t epochs for the training set. For example in Fig. 4(g), at timestamp 3, the prediction accuracy of SVM classification is 0.95: this value shows the performance-level prediction after observing 3 training epochs for the ResNet architecture. The signatures in both the training and test set are constructed based on only those 3 epochs during the training process.

---

### Official Review · Reviewer_Txru · 2021-11-05

**Correctness:** 4
**Technical Novelty And Significance:** 3
**Empirical Novelty And Significance:** 2
**Recommendation:** 6
**Confidence:** 3

**Main Review:**

The graph analysis perspective for understanding neural networks is definitely timely and promising. The proposed method seems (experimentally) efficient in terms of the number of epochs and prediction performance. Moreover, in comparison to state of the art methods such as Rieck et al.'19, the methods is computationally more efficient.

There are however different points that should be clarified in the text.

1) The whole framework should be better explained. While Fig. 1 is very useful, the paper is missing a connection between Fig. 1 and the graph construction (Figs. 3,4). In particular, it would be useful to have a picture that shows the graph structure for a 'simple' deep neural network, with only a few filters and layers, as well as the number of edges corresponding to each step. Fig. 3 is good, but it does not give the complete picture to the reader.

2) The authors should explain a bit more the intuition behind these graph signatures. What type of dynamics do they capture? What do they reveal about the architecture? What is the advantage in comparison to other graph features (local or global)?

3) The authors claim that the method can be used in interpretability-related tasks. Can you elaborate on that or give some examples?

4) How do you define the training set for a specific architecture? For one architecture, and one dataset, you have a single time-varying graphs, no?

4) The paper needs some work in terms of presentation. There are different typos or expressions that should be clarified. Some of them are listed below:
   - pg.3 "results feature-map" --> "results in feature-map"
   - pg.3 "number of nodes" --> "the number of nodes"
   - pg.3 "layer of graph" --> "layer of the graph" (the entire sentence needs to be rewritten)
   - pg.4 Fig.3: K layers means K node types? Does the type of the node play a role in defining the features?
   - pg.4: section 3.2: "the introduced time-evolving graph representations" . However, from my understanding, no time-varying graph representation has been introduced till that point.
   - pg.6: Please clarify what is the binary classification in Table 2
   - pg.7: "Imagent" --> "ImageNet"
   - Please explain the green colour in Fig. 6.
   - pg.8: "changes of rolled graph graph" --> "changes of the rolled graph"
   - pg.8: "a few epochs prediction model shows" --> "a few epochs, the prediction model shows"
   - pg.8: "that needed for" --> "that were needed for"
   - pg.9: Limitations: please check different typos in that paragraph
   - Please make references consistent; some of them are even missing the venue (e.g., Krizhevsky et al.) or page numbers.
   - pg.12: 'results the poorest accuracy" --> "result the poorest accuracy"

**Summary Of The Paper:**

This paper tries to model the learning dynamics of a (deep) neural network using graphs. In particular, the authors propose to represent the learning process as a time-varying graph, which is then used to capture the structural changes through some simple graph statistics (such as weighted degree and eigenvector centrality), and eventually to predict the accuracy of the neural network (through some simple ML models such as SVMs or MLP).

**Summary Of The Review:**

The paper still needs some work to merit a publication. However, I would be willing the accept the paper if the authors do a careful revision and address all the comments raised by the reviewers.

---

> ### Author Response · Authors · 2021-11-23
> **Response to reviewer Txru part 1**
>
> Thank you very much for your valuable feedback. We would like to clarify some points:
>
> 1- “Improving Figure 3”
>
> Thank you for your recommendation. We updated Figure 3 to illustrate a NN with two convolutional layers and one fully connected layer on the left; on the right, we show the graph representation for that NN. Note that, the number of edges between each two layers (i.e., layer i and layer j) of the represented graph is less than or equal to <#nodes in layer_i> x <#nodes in layer j> due to dropout; some edges would be dropped from the graph and the corresponding edge weight would be zero.
>
>
> 2- “Intuition behind the graph signatures”
>
> Based on your feedback, we have updated our paper (Sec. 3.2 (S2) and (S3)) with additional justifications and intuition behind the graph features that we leverage during the signature construction step (degree and eigenvector centrality).
> We chose weighted degree centrality to efficiently capture (1) the _weights_ that are learned in each layer of the NN at the node level (filter or neuron), which are the most important aspect of the training process;  and (2) the _change_ of the learned weights during the training process of NN over time (in the proposed time-evolving graph representation). Moreover, we use the node degrees to model the NN structure since they have been found to be effective in capturing the structural roles of nodes and a key notion for structural node embedding [1]. We note that we show the difference in how the degree-based signatures change during the training process for low accuracy NNs vs. high accuracy ones in Figure-7-9 in the appendix.
>
> Eigenvector centrality is used to capture the highly influential nodes in a graph; in our case, the nodes correspond to filters of convolutional layers and neurons of fully connected layers. Additionally, eigenvector centrality has been used for detecting communities [2] or clusters [3], and thus provides structural information about the clusterability of the NN. Clusters or communities are central for network analysis and provide complementary information compared to the simpler and more efficient-to-compute degree centrality. Our hypothesis was that tracking changes in the eigenvector centrality (or node influence or clustering-related information) over time could provide complementary insights into the structural changes of the NN during training that may correlate with high or low performance.
> *Global vs. local properties*: Our proposed framework requires global graph summaries in order to be able to compare different NN-based graphs (with different number of nodes and edges) in the training set; we achieved this by computing statistical summaries (signatures) of graph properties that capture the structure locally at the node level and are efficient to compute (so that our approach adds little overhead). We also experimented with other global features such as algebraic connectivity, modularity, and average shortest paths, but those fail to capture the structural changes in our graphs and lead to poor classification performance (they remain constant at different timestamps of the represented graphs).
> *Why not use other graph properties?* Our proposed graph representation is k-partite, and thus many commonly used graph features are not appropriate.  For example, clustering-based features (such as number of triangles, transitivity, clustering coefficient, square clustering coefficient), and cycle-based metrics which account for closed paths in a graph (e.g., cycle basis, minimum cycle basis) are always equal to 0 due to the k-partite structure of our graphs, and so fail to model the graph structure. Connected component-related metrics (i.e., connectivity, strong connectivity, weak connectivity) do not capture the weighted edges (i.e., the learned weights), which are important in our scenario for modeling the training dynamics of NNs. In terms of other centralities, betweenness centrality (which is commonly used in network analysis) is based on all the shortest paths that pass through a particular node, which makes it computationally expensive and less efficient than eigenvector centrality.
> [1] Jin, J. et al. “Toward Understanding and Evaluating Structural Node Embeddings”. ACM Transactions on Knowledge Discovery from Data 2022. arXiv:2101.05730v1
> [2] Newman, Mark EJ. "Finding community structure in networks using the eigenvectors of matrices." Physical review E 74.3 (2006): 036104.
> [3] Wu, Qin, et al. "“follow the leader”: A centrality guided clustering and its application to social network analysis." The Scientific World Journal 2013 (2013).

---

> > ### Author Response · Authors · 2021-11-23
> > **Response to reviewer Txru part 2**
> >
> > 3- “Interpretability-related tasks”
> >
> > Thank you for this question. We have updated our paper to phrase our statements regarding interpretability more clearly (Section 3.1 and Section 5); we realize that our previous claims could be confused with “NN interpretability”, a topic that has a different focus than our work.
> >
> > Our main claim is that our proposed rolled graph representation is more interpretable than the unrolled representation---i.e., it is easier to understand what it represents. In our proposed rolled graph representation, the nodes are defined to represent filters in convolutional layers; on the other hand, the nodes represent feature maps in the unrolled representation introduced by (Rieck et al, 2019). Thus, we argue that it is easier to interpret the outputs of downstream graph analysis on our graph representation (e.g., computing node properties, tracking the evolution of the graph, performing community detection). For example, when we calculate the eigenvector centrality, it is clear which filters (from a specific layer) are the most influential filters; if we were to do the same calculation on the previously introduced unrolled representation, it would not be possible to draw such a conclusion. Furthemore, we believe that our graph representation can be a powerful tool for future studies that aim to understand the NN dynamics.
> >
> >
> > 4- “Training set for a specific architecture”
> >
> > Due to the space limitation we moved the experimental configuration to Appendix A.1. For each architecture and dataset, we train 480 NNs (48 learning rates and 10 dropout values) and then we construct the time-evolving graphs corresponding to those different NN configurations.
> >
> >
> > 5- “Typos”
> >
> > We really appreciate your detailed review of our paper; we carefully edited all the typos and addressed the presentation issues.

---

> > > ### Comment · Reviewer_Txru · 2021-11-29
> > > **After revision**
> > >
> > > I would like to thank the authors for their detailed responses. I feel that after the revision, the quality of the paper has improved. Thus, I am willing to increase my score to 6. While the work is mainly experimental, without clear theoretical support, I find the direction promising, and worth exploring further.

---

### Official Review · Reviewer_cuEB · 2021-11-08

**Correctness:** 4
**Technical Novelty And Significance:** 3
**Empirical Novelty And Significance:** 3
**Recommendation:** 8
**Confidence:** 3

**Main Review:**

* Strenghts
- The paper is rather well written, clear and coherent (I could only spot a typo “ changes of rolled graph graph between”,  with the repeated “graph” word)
- The empirical analysis is rather extensive and the results seem to support the claims
* Weaknesses
- Lack of theoretical insights makes the contribution slightly less relevant and appealing
- Lack of asymptotic computational (time and space) complexity analysis would definitely add some more value to the paper, especially when comparing the behavior wrt early stopping and the literature alternatives.


**Summary Of The Paper:**

The paper proposes a graph-based approach to assess the performance of deep neural networks in vision tasks. The approach is essentially based on representing the first few epochs of SGD-based training of the neural architecture seen as a graph (where nodes stand for neurons and edges stand for connections among neurons), followed by feature extraction, signature construction and finally a classifier (to predict the high or low level of final accuracy) or a regress or (to predict the exact finally accuracy score) layer.
The approach is assessed on several experimental settings, though limited to the case of static networks (e.g., without recurrent/recursive connections). The results seem encouraging and interesting

**Summary Of The Review:**

The paper is overall convincing, even though lack of theoretical insights and of computational complexity analysis reduce a bit my final evaluation.

---

> ### Author Response · Authors · 2021-11-23
> **Response to reviewer  cuEB**
>
> Thank you for your time and feedback. We are glad that you found our methodology convincing.
>
> Based on your suggestion, we updated the paper and added the time complexity of our proposed rolled graph representation in Section 4.3.  We also describe the size of generated graphs in Section 3.1.

---

### Author Response · Authors · 2021-11-23
**General response part 1**

We thank the reviewers for their constructive feedback. We are glad that the reviewers liked various aspects of our paper, such as the convincing extensive empirical analysis (cuEB), the technical soundness of our paper (zP52), and the potential of our proposed graph representation for understanding NNs (Txru). In our general response, we would like to emphasize the motivation and contributions of our work, and the intuition behind the graph signatures in our proposed method. We address the questions raised by each reviewer in our individual responses.

Motivation & Contributions
This problem is inspired by the growing interest in explainable AI and the need to better understand the widely-used artificial neural networks. Towards this end, we introduced a new graph representation that can model a variety of popular NNs and their learning dynamics during the training process in an intuitive way. Unlike prior work that focuses on static representations or other abstractions, our representation is time-evolving and its components (nodes, edges), which capture the learned parameters (by design), are easy-to-understand. To show the effectiveness of our graph representation, we formulated the performance prediction task, which investigates the relation between the structural changes in our graph-based NN representations and the NN outcomes (performance) for different hyperparameter configurations; we predict both the actual value of accuracy for an NN model and its performance level (high/low accuracy). We believe that the empirical results are promising and highlight the value of our graph model, which could open up new research directions towards understanding the NN structure and its interplay with other factors.


Intuition behind our graph signatures [Reviewers <Txru>, <DNSD>]

We have updated our paper (Sec. 3.2 (S2) and (S3)) with additional justifications and intuition behind the graph features that we leverage during the signature construction step (degree and eigenvector centrality).
We chose weighted degree centrality to efficiently capture (1) the _weights_ that are learned in each layer of the NN at the node level (filter or neuron), which are the most important aspect of the training process;  and (2) the _change_ of the learned weights during the training process of NN over time (in the proposed time-evolving graph representation). Moreover, we use the node degrees to model the NN structure since they have been found to be effective in capturing the structural roles of nodes and a key notion for structural node embedding [1]. We note that we show the difference in how the degree-based signatures change during the training process for low accuracy NNs vs. high accuracy ones in Figures 7-9 in the appendix.
Eigenvector centrality is used to capture the highly influential nodes in a graph; in our case, the nodes correspond to filters of convolutional layers and neurons of fully connected layers. Additionally, eigenvector centrality has been used for detecting communities [2] or clusters [3], and thus provides structural information about the clusterability of the NN. Clusters or communities are central for network analysis and provide complementary information compared to the simpler and more efficient-to-compute degree centrality. Our hypothesis was that tracking changes in the eigenvector centrality (or node influence or clustering-related information) over time could provide complementary insights into the structural changes of the NN during training that may correlate with high or low performance.

*Global vs. local properties*:

Our proposed framework requires global graph summaries in order to be able to compare different NN-based graphs (with different number of nodes and edges) in the training set; we achieved this by computing statistical summaries (signatures) of graph properties that capture the structure locally at the node level and are efficient to compute (so that our approach adds little overhead). We also experimented with other global features such as algebraic connectivity, modularity, and average shortest paths, but those fail to capture the structural changes in our graphs and lead to poor classification performance (they remain constant at different timestamps of the represented graphs).

---

### Author Response · Authors · 2021-11-23
**General response part 2**

*Why not use other graph properties?* Our proposed graph representation is k-partite, and thus many commonly used graph features are not appropriate.  For example, clustering-based features (such as number of triangles, transitivity, clustering coefficient, square clustering coefficient), and cycle-based metrics which account for closed paths in a graph (e.g., cycle basis, minimum cycle basis) are always equal to 0 due to the k-partite structure of our graphs, and so fail to model the graph structure. Connected component-related metrics (i.e., connectivity, strong connectivity, weak connectivity) do not capture the weighted edges (i.e., the learned weights), which are important in our scenario for modeling the training dynamics of NNs. In terms of other centralities, betweenness centrality (which is commonly used in network analysis) is based on all the shortest paths that pass through a particular node, which makes it computationally expensive and less efficient than eigenvector centrality.
[1] Jin, J. et al. “Toward Understanding and Evaluating Structural Node Embeddings”. ACM Transactions on Knowledge Discovery from Data 2022. arXiv:2101.05730v1
[2] Newman, Mark EJ. "Finding community structure in networks using the eigenvectors of matrices." Physical review E 74.3 (2006): 036104.
[3] Wu, Qin, et al. "“follow the leader”: A centrality guided clustering and its application to social network analysis." The Scientific World Journal 2013 (2013).

---

### Author Response · Authors · 2021-11-29
**A kind reminder to the reviewers**

Since today is the last day for the discussion, can you please take a look at our responses and let us know if we have sufficiently addressed the points that you raised in your original reviews and if you have any further questions?

---

### Decision · Program_Chairs · 2022-01-20

**Decision:**

Reject

**Comment:**

This paper proposes to represent a deep neural network as a graph and analyze its learning dynamics as a time series of weighted graphs corresponding to the neural network. As the graph representations, the authors propose to use a rolled representation in addition to a unrolled representation. Then, they proposed to utilize the graph features of the representations for predicting its predictive accuracy.

This paper presents an interesting idea which could be used for predicting the test accuracy from the first few epochs training. However, there are also several weaknesses as pointed out by the reviewers. First, the justification of using the graph structure to predict the accuracy is weak (indeed, the graph structure can be used for prediction, but its necessity is not well supported), and there is no theory to support the proposed method. Second, the problem setting is a bit wired. The training data is generated by using the same architecture and data set. Although the authors gave additional experiments on the architecture generalization, it is still difficult to see how convincing the method is for more general settings. Third, baseline methods are not shown in their experiments.
In addition to that, the thresholds for the classification tasks seem to be too small (like 40% in CIFAR10) which would make the problem too easy. Therefore, the practicality of the method is rather unclear.

This paper is quite on the borderline, but for the reasons listed above, it is a bit below the acceptance threshold.